# Perceptions and Practices of Oral Health Care Professionals in Preventing and Managing Childhood Obesity

**DOI:** 10.3390/nu14091809

**Published:** 2022-04-26

**Authors:** Amit Arora, Kritika Rana, Narendar Manohar, Li Li, Sameer Bhole, Ritesh Chimoriya

**Affiliations:** 1School of Health Sciences, Western Sydney University, Locked Bag 1797, Penrith, NSW 2751, Australia; k.rana@westernsydney.edu.au (K.R.); drnarendar@gmail.com (N.M.); r.chimoriya@westernsydney.edu.au (R.C.); 2Health Equity Laboratory, Campbelltown, NSW 2560, Australia; 3Translational Health Research Institute, Western Sydney University, Locked Bag 1797, Penrith, NSW 2751, Australia; 4Discipline of Child and Adolescent Health, The Children’s Hospital at Westmead Clinical School, Faculty of Medicine and Health, The University of Sydney, Westmead, NSW 2145, Australia; 5Oral Health Services, Sydney Local Health District and Sydney Dental Hospital, NSW Health, Surry Hills, NSW 2010, Australia; sameer.bhole@health.nsw.gov.au; 6Australian College of Physical Education, Sydney Olympic Park, NSW 2127, Australia; 7School of Science, Western Sydney University, Locked Bag 1797, Penrith, NSW 2751, Australia; l.li7@westernsydney.edu.au; 8Sydney Dental School, Faculty of Medicine and Health, The University of Sydney, Surry Hills, NSW 2010, Australia; 9School of Medicine, Western Sydney University, Campbelltown, NSW 2560, Australia

**Keywords:** oral health care professionals, childhood obesity, prevention, management, qualitative study

## Abstract

In this study, we aimed to explore the perceptions of oral health care professionals (OHCPs) on childhood overweight and obesity screening and management in oral health settings in the Greater Sydney region in New South Wales, Australia. OHCPs involved in the Healthy Smiles Healthy Kids (HSHK) birth cohort study were purposively selected for this nested qualitative study. A sample of 15 OHCPs completed the face-to-face interviews, and thematic analysis was undertaken to identify and analyse the contextual patterns and themes. Three major themes emerged: (1) obesity prevention and management in dental practice; (2) barriers and enablers to obesity prevention and management in dental settings; and (3) the role of oral health professionals in promoting healthy weight status. This study found that OHCPs are well-positioned and supportive in undertaking obesity screening and management in their routine clinical practice. However, their practices are limited due to barriers such as time constraints, limited knowledge, and limited referral pathways. Strategies including capacity building of OHCPs, development of appropriate training programs and resources, and identification of a clear specialist referral pathway are needed to address the current barriers. This study provides an insight into opportunities for the oral health workforce in promoting healthy weight status among children.

## 1. Introduction

Childhood overweight and obesity is a global public health concern, and its prevalence has increased dramatically over the past few decades [1]. Overweight and obesity during childhood is associated with greater risk of developing chronic diseases such as cardiovascular diseases, diabetes, and non-alcoholic fatty liver disease at a younger age [2,3]. Childhood overweight and obesity is associated with increased risk of obesity in adulthood, and it predisposes children to several non-communicable diseases and psychosocial issues in later life [1,2,4,5,6].

One of the most common chronic diseases globally, dental caries (tooth decay) also continues to present an important public health challenge for children [7,8]. Untreated dental caries may lead to severe pain, loss of sleep, and poor nutrition, thus compromising growth and development in children [9,10]. Decay in primary (baby) teeth is also associated with increased risk of dental caries in the permanent dentition [11]. Furthermore, oral disorders including dental caries in children have been a major contributor to the global disease burden for the past three decades [12].

In Australia, the prevalence of overweight and obesity in children has steadily increased from 20% in 1995 to 25% in 2017–2018 [4], and dental caries has remained a major contributor of disease burden among children [13]. It is suggested that overweight or obesity and dental caries are associated and share common aetiological factors, such as high-sugar diet, limited use of health services, and low socio-economic status [7,14]. Recent systematic reviews have also identified some evidence on the possible link between childhood obesity and dental caries [7,15,16]. Nonetheless, epidemiological evidence highlights significant challenges in curtailing the burden of overweight and obesity and dental caries in children [7]. The World Health Organization (WHO) recommends using the common-risk factor approach (CRFA) to reduce the burden of oral diseases, which focuses on shared risk factors to address different health problems including childhood overweight or obesity [17]. Therefore, there is a need to design interventions targeting aetiological factors commonly shared between overweight or obesity and dental caries, thus addressing both issues concurrently [14].

Integrating oral health with general health promotion is now recommended in several countries, which is also supported by the WHO recommendations [17,18]. Australia’s National Oral Health Plan 2015–2024 has listed oral health to be an integral part of general health promotion [19]. While England’s health policies on reducing sugar consumption among children are mainly focused on childhood obesity prevention, a substantial proportion of dental caries cases are projected to be avoided because of these policies [20]. The CRFA in oral health promotion rationalises the concept of developing and implementing childhood overweight and obesity interventions that can be integrated in oral health settings [21]. A few prior studies also support the need for addressing childhood overweight and obesity in the dentistry setting [22,23]. Providing brief targeted nutritional screening and/or dietary advice to promote both oral health and healthy weight as part of routine dental practice can further enhance oral health systems to be an integral part of primary health care [17]. 

A recent review highlighted the potential role of oral health care professionals (OHCPs) in childhood overweight and obesity management [24]. Possible weight management strategies include height and weight measurements, nutritional screening and counselling, and referrals to weight management services [24]. Moreover, there seems to be high acceptability and feasibility of such service integration, particularly by dental hygienists, oral health therapists, and paediatric dentists [24], with some dental services already adopting relevant screening, assessment, and counselling procedures as part of their oral health service [25,26]. Parental acceptance towards healthy-weight-related services in oral health settings has also been previously reported [27]. 

Provision of relevant training to OHCPs is needed to build their capacity to support strategies, advocated by the WHO, and beneficial for both oral health and healthy weight management [17]. Developing and delivering formal training related to weight management and nutrition counselling to the OHCPs will help provide a more holistic health care experience to paediatric clients at risk of overweight and obesity. There is also a need for peak bodies of dental practitioners, weight management specialists, nutritionists, and dietitians to redefine the scope of practice to create formal referral pathways and minimise service overlap. Relevant policies and regulations can then be planned to efficiently promote healthy weight and oral health in children [24]. 

Global evidence to support the integration of childhood overweight and obesity management in dental care settings is available [22,23]. However, relevant information on the acceptance, feasibility, enablers, and barriers to such practices amongst Australian OHCPs is scarce. Hence, in this study, we aimed to explore the perceptions of OHCPs on childhood overweight and obesity screening and management in oral health settings in the Greater Sydney region in New South Wales (NSW), Australia. This study will aid in identifying the readiness and limitations of current practices of OHCPs in adopting weight screening and management. It will also inform the development of a strategic plan to enable this service innovation and integration.

## 2. Methods

### 2.1. Study Background

This qualitative study is nested within the Healthy Smiles Healthy Kids (HSHK) study, which is a large ongoing multi-centre birth cohort study in the Greater Western Sydney region in NSW, Australia [10,28]. The HSHK study was commenced in 2010 to examine the relationship between early childhood feeding patterns, oral health, and obesity [29]. The details on the HSHK study have been provided elsewhere [30,31]. As part of the HSHK study, this qualitative study was conducted with OHCPs involved in the HSHK study as participants. 

### 2.2. Study Design

In this study, a qualitative research design was employed to gain a comprehensive understanding of OHCPs’ views on obesity screening and management in dental settings [32]. The qualitative research method is a frequently used approach to explore complex social phenomena as well as to collect in-depth information on health professionals’ experiences, views, and understanding on a wide range of topics [33,34,35]. The flexibility of the qualitative research design also allowed for simultaneous data collection and data analysis, as well as further investigation and analysis of the emerging themes [33]. 

### 2.3. Sampling

A purposive sampling technique was used to recruit OHCPs for this nested qualitative study [36]. A non-probability sampling method, purposive sampling allows for deliberate selection of specific individuals to acquire information-rich data [37]. The sample size was determined based on data saturation. The recruitment of participants continued until all dimensions of interest were explored and the addition of new participants did not aid in generating any new information that would enhance the study findings [33]. 

A total of 15 OHCPs involved in the HSHK study were invited to participate in this study through a telephone call. The OHCPs included dental therapists, oral health therapists, general dentists, and paediatric dentists. The OHCPs were sent an information pack containing a participant information sheet and a consent form via email prior to the interview. All 15 OHCPs completed the interviews, and this was sufficient to reach data saturation, where no new topics or information emerged from the interviews. Previous studies have also found that a total of 12–15 participants is adequate to achieve data saturation [37].

### 2.4. Semi-Structured Interviews

In-depth semi-structured interviews were conducted in 2018 and 2019 by two researchers (A.A. and N.M.) with extensive experience in population oral health and qualitative research. Face-to-face interviews were conducted for approximately one hour. A semi-structured interview guide (Appendix A) was utilised to facilitate exploration of several related topics for discussion in the interviews. The interview guide was developed based on the key areas of interest identified through comprehensive systematic literature reviews [7,24]. The semi-structured interview process allowed participants to speak freely, and the interviewers probed the participants, where appropriate, to facilitate discussions on any topic outside the interview guide. All interviews were audio-recorded, debriefed immediately, and transcribed verbatim by employing a professional transcription service. 

### 2.5. Data Analysis

Thematic analysis involving an inductive approach was undertaken to identify and analyse the contextual patterns and themes within the data gathered from the interview transcripts. The five steps of thematic analysis were followed: data familiarisation, generation of initial codes, searching for themes, review of themes, and defining and naming themes [38]. The transcripts were checked for accuracy and imported into Quirkos (Quirkos, Edinburgh, Scotland, UK), a qualitative data management and analysis software. The transcripts were read and re-read individually to gain familiarity with the data. Using the Quirkos software, one researcher (K.R.) performed the initial coding and identified the common themes and subthemes. Three researchers (A.A., R.C. and L.L.) independently reviewed the data for manual coding and analysis and identified the underlying concepts. All four researchers compared and reviewed the results of independent manual coding and coding in Quirkos, and where possible, the coding was merged to form themes and subthemes. Any inconsistencies were resolved through open discussions between all four researchers to reach a unanimous decision. 

### 2.6. Rigor

Numerous methodological strategies were adopted to enhance the rigor of this study. The interviews were conducted by two researchers with extensive experience in population oral health and qualitative research. Interview debriefings were consistently conducted until the data saturation was confirmed. Following each interview, the data collected were reviewed, the completeness of the data was checked, the key findings were identified, and any further topics to explore in the subsequent interviews were determined. A professional transcription service was employed to enhance the accuracy of the verbatim transcriptions of the audio recordings. The interview transcripts were shared with the participants to confirm the accuracy. During data analysis, four researchers independently performed coding, and consensus was achieved between all researchers. Direct quotes were identified to support the key themes and subthemes and are presented in the results. Moreover, adequate information on the study setting, study participants, and data collection and analysis methods have been provided. The criteria for robust qualitative research, including credibility, transferability, dependability, and confirmability, have been addressed in this study with the use of these methodological strategies [39,40].

### 2.7. Ethical Considerations

The ethical approval for the HSHK birth cohort study was acquired from the Human Research Ethics Committee of the former Sydney South West Area Health Service—RPAH Zone (ID number X08-0115), Western Sydney University, and the University of Sydney. This study was conducted according to the guidelines of the World Medical Association Declaration of Helsinki. Written informed consent was obtained from all participants.

## 3. Results

Each face-to-face interview lasted between 45 and 75 min. Of the 15 OHCPs, 5 worked in South-Western Sydney, 4 in Inner-West Sydney, and 6 in the Western Sydney region of NSW, Australia. Of the 15 OHCPs interviewed in this study, 9 were dental therapists or oral health therapists, 4 were general dentists, and 2 were paediatric dentists. A total of 11 participants worked solely in public oral health services and 4 worked in both public service and private practice. Of the 15 participants, 11 were female and 6 were over the age of 30 years (range 26–60 years). The mean time that the OHCPs had worked clinically was 14 years (range 5–30 years). 

The thematic analysis of the interview data generated three major themes: (1) obesity prevention and management in dental practice; (2) barriers and enablers to obesity prevention and management in dental settings; and (3) the role of oral health professionals in promoting healthy weight status. The themes and subthemes are presented in Table 1.

### 3.1. Theme 1: Obesity Prevention and Management in Dental Practice

#### 3.1.1. Perceived Link of Oral Health and General Health

Most OHCPs were knowledgeable of the link between oral health and general health. All participants were successful in linking oral health and obesity through diet, whilst they considered the link between gum disease and diabetes was more established. They also conveyed the concepts of the association to their patients:

“*Gum disease and diabetes or tooth decay and obesity, they are definitely linked. It’s one of the things that I talk about a lot in the clinic, as well as going on about how to brush your teeth. I explain that it’s important and healthy teeth are also part of a healthy body*.”

Some OHCPs elaborated on the strong link between dental caries and obesity, attributing this to the high intake of cariogenic diet. Some participants highlighted that those with obesity are at a high risk of having dental caries, which elucidates the link between the two conditions:

“*For any food, the first direct contact is inside the mouth. So, most patients who are obese, they have a high intake of cariogenic (high sugar) diet. And because of that, they have a higher rate of incidence of dental caries. Usually, I find that patients with obesity have a high risk of getting dental caries than others that are not obese*.”

#### 3.1.2. Understanding of the Common Risk Factors

**Diet:** All OHCPs identified sub-optimal diet, particularly junk food and high-sugar food and beverages, as a major contributor to both weight gain and dental caries:

“*These days there’s a lot of junk food and high sugar foods and beverages that contribute to weight gain, and same goes for caries. So, we do see more children having decayed teeth if they are exposed to more sugar and junk food. As a result, kids put on the weight from eating these foods*.”

Although food was commonly linked with obesity and dental caries, some OHCPs suggested that food may contribute more to dental caries than obesity, considering other contributing factors to obesity such as exercise and genetics:

“*It’s just the food that is common, there is definitely a link between obesity and caries. It is more the foods that these children are eating that is contributing to both obesity and caries, at similar rates and linking more to caries. With obesity, we can definitely say there is a link there, but there are other factors that we have to account for, like exercise and genetics*.”

**Socioeconomic status:** Most OHCPs associated low socioeconomic status with sub-optimal diet, which in turn contributes to both obesity and oral diseases. Some participants emphasised that junk food and sugar-sweetened beverages are exceptionally cheap, and mostly even cheaper than water, which may increase the consumption of these foods among those from low-income families:

“*Part of it is the socioeconomic in the area that you’re in. Some of them tend to snack a lot on chips and chocolate because the sweet junk food is so cheap. Slurpee is one dollar, and you go to the movies, a bottle of water is five dollars. The sweet drinks are way cheaper than water. They should put limits on how much they charge for water really everywhere*.”

A few OHCPs who worked in a low socioeconomic area underlined that as healthier foods are often more expensive than fat foods, it is a challenge to persuade patients to adopt a healthier diet in order to reduce the risk of both dental caries and obesity:

“*The local health district I’m working, a lot of people are low socioeconomic. So, it’s very hard to get them to change their diet. Because anything that is healthy is more expensive than fat foods. Especially in our region, as they are all low socioeconomic, the choice for them is not much. They’re more happy to get junk food than to get healthy foods*.”

#### 3.1.3. Current Prevention and Management Practices

**Diet assessment:** The majority of the OHCPs were undertaking diet assessments informally through questions about the child’s diet and without using any diet diary or food frequency questionnaire. Most participants suggested the diet assessment was primarily focused on the child’s behaviour when it came to their diet, their eating patterns, and especially the consumption of sugar-sweetened beverages, fruit juices, breakfast meals, treats, and snacks. As per the child’s needs, some OHCPs also suggested adding vegetables to their diet:

“*I usually ask the mom first, how is their diet? Are they fussy eater? I direct my talk on what the patient needs. If the patient is fussy then guiding the parents to incorporate a little bit of veggies, keep the treats and fruit juices to a minimum. I use that sugar chart on the sweet beverages and breakfast food, and snacks*.”

Some OHCPs indicated that they took the Australian Dietary Guidelines as a reference to advise on increasing the consumption of fruits and vegetables, as well as the sampling size. Based on the diet assessment, some participants also suggested healthier alternatives to the sugary foods and beverages the children were consuming:

“*We talk about the Australian Dietary Guidelines, making sure they’re having their fruits, two fruits and five veggies a day or sampling size. Basically, water over sugary drinks, in particular tap water because of the fluorides. We do talk about diet, when it comes to sugary snacks and foods, we try to find some healthy alternative to snacks, cheese, fruit sticks, hummus dips, if they’re not allergic. So, just looking at healthier alternatives*.”

**Patient education:** All OHCPs were providing patient education in the form of dietary advice in order to prevent or manage both obesity and dental caries:

“*I tell them that because of obesity, and the diet that the patient is having, it is causing a little problem with our health. If you can cut down the diet, then you have less decay, you’ll be less overweight. So, there is a win-win situation, when you control this sort of thing*.”

The OHCPs suggested that the dietary advice they provided was primarily focused on limiting the intake and frequency of sugars, snacks, and beverages, and avoiding sticky sugar-containing foods. Some participants highlighted that they generally tried to be understanding and provided tips on limiting sweets and juice intake:

“*We do try and be a little bit more understanding. We try to explain when the best time is to have something sweet, like during a mealtime. And if they do have things like juice, we recommend watering it down so it’s not as concentrated, and to have it with a straw. So, we do give them like tips like that*.”

**Providing referrals to dietitians and general practitioners:** The majority of the OHCPs outlined that despite the time constraints, they would provide dietary advice to their patients who were overweight. Nonetheless, they were providing referrals to dietitians and general practitioners (GPs), especially for the management of obesity:

“*Unfortunately, our appointments are not as long as we wish they would be. If there was a patient that had some issue with their weight, I would suggest reducing sugar, reducing portion size, and increasing water. But if there is a big issue, I recommend a referral to a dietitian or the general practitioner.*”

Most OHCPs indicated that there were no formal referral processes available, and only children who were overweight or had obesity were referred to either a GP or dietitian:

“*So, I have referred a patient who was overweight to a GP or dietitian because I don’t really know of any other places that that they can go to*.”

### 3.2. Theme 2: Barriers and Enablers to Obesity Prevention and Management in Dental Settings

#### 3.2.1. Barriers for Oral Health Professionals

**Time:** Time constraints on the part of OHCPs were considered to be major barriers. Although an appointment lasted 40 min, the OHCPs emphasised that it would not be sufficient to provide counselling on obesity prevention and management:

“*It is 40 min ideally for a first appointment. If you have a child that’s in the chair that is crying, mom just wants to get out of there. Even if you have a topic that is of concern and want to talk about that, some parents just want to leave, because the child is acting up or they have got some other commitments*.”

Especially in a 40-min pain relief appointment, time constraint was elucidated as a major barrier to undertaking anthropometric assessment and weight counselling:

“*Time would be an issue given the time constraints. Especially if they’ve come in for a pain appointment and you have to do the height and weight measurement. And you might not have enough time to treat their pain if it goes into depth with the talk about the diet*.”

**Referral pathways:** The OHCPs identified uncertainty about the referral pathways as a key barrier to obesity prevention and management. Most participants mentioned that they were not referring their patients to specialists due to the lack of knowledge regarding the referral pathway process. Nonetheless, they were providing referrals to GPs:

“*In case someone comes with weight issues, I don’t refer them to specialists because I don’t know how to. I would just tell them you go to your GP, who will answer all your questions. Usually, GPs have dietician and nutritionist in their practice so we can incorporate them*.”

The OHCPs noted the importance of integrated care and highlighted the need for OHCPs, nutritionists, and GPs to work together. Most OHCPs also suggested the need for clear referral pathways and communication between health care professionals regarding the prevention and management of obesity:

“*There will have to be referrals in place. If there is an issue, you can bring it up and they can correspond with a letter saying when and what treatment they’ve provided the patient with. If there’s something more serious, the specialists would be the one to go and then they would probably refer to someone that can help with the preventative side of things*.”

**Limited knowledge on dietary guidelines recommendations:** The OHCPs in this study agreed that they had limited knowledge on the Australian Dietary Guidelines recommendations. Some OHCPs indicated that they used the Australian Dietary Guidelines as a reference to provide dietary advice, which was restricted to limiting the frequency of consumption of sugary and sticky foods and having five vegetables and two fruits a day. However, they were not aware of the recommended age for a baby’s first foods, and that foods with added sugar should be discouraged in the first 12 months. Most participants suggested that although they were not aware of the Australian Dietary Guidelines recommendations, the dietary advice they provide could have stemmed from the guidelines:

“*We give dietary advice but don’t know what the Australian Dietary Guidelines recommend. The advice that we give currently is just based on experience. And it probably stemmed from guidelines. I would have had some knowledge and just not realise it’s on the guidelines*.”

Some OHCPs highlighted that they were unsure about the dietary advice they were providing to the children and families, and identified the need to get acquainted with the Australian Dietary Guidelines recommendations in order to provide accurate advice:

“*I don’t want to give false information. I need to do some study to see what proper dietary guideline is. At the moment, I’m not too sure what the best advice is for the patient. I’d like to read the whole guideline and see what it is for specific age group*.”

**Limited knowledge on Body Mass Index (BMI), BMI for age chart, and obesity:** Most OHCPs mentioned that they had limited knowledge on both BMI and obesity. Many OHCPs were unaware of the BMI cut-off points for obesity and emphasised the need to learn about both:

“*I would like to know more because I don’t know much about BMI and obesity. All I know is that a high BMI means they could be overweight. But I don’t know about the cut-off points and when it is normal. I need to know everything from scratch.*”

All OHCPs mentioned that they were not undertaking BMI charting, especially because of the limited knowledge on the BMI for age chart:

“*I don’t know the cut-off points for obese or overweight. There is a chart on this, I don’t know enough for the ages though. BMI should be different for adult and children and depending on the age. The different charts, the different ways BMI is measured, I definitely want to learn that. I’m not doing the charting of the BMI at the moment*.”

**Scope of practice:** Uncertainty towards scope of obesity screening and counselling was identified as a significant barrier by the OHCPs. A few participants suggested that other health professionals would be more qualified to undertake obesity screening:

“*Sometimes you feel a bit uncomfortable measuring the height, weight and waist because we’ve got other health professionals that might be more qualified, like dietitians and GPs. Especially with children, we weren’t so focused on their height or weight, and more focused on their teeth, diet, and oral hygiene habits*.”

Some OHCPs suggested that uncertainty of scope of practice is a greater barrier in private practice. Some participants emphasised that even if obesity screening and counselling were charged extra in a private practice, they would not be certain about providing the service:

“*When you’re working in the clinic, you want to do everything to the best of your ability. But when you’re in private practice, there’s also financial backing that you need to consider. Diet advice or oral hygiene don’t necessarily attract a dollar figure. I think that’s a barrier for a clinician to provide that service. Even if there was an item code that attracted a dollar figure, you can’t guarantee that they’ve done diet advice or taken weight or height*.”

**Communicating advice to children and families:** Most OHCPs reported hesitation to initiate a discussion around children’s weight status with parents as an additional obstacle. Some participants expressed a fear of offending the children and families or appearing judgemental when communicating advice about obesity as a dental practitioner:

“*You get worried if you say something that might offend the mother or the child. Maybe somebody before you hasn’t picked up on anything, then, parents might think why a dental practitioner is worried about the weight or the height, having gone to a nurse or another health care professional.*”

Nonetheless, some OHCPs suggested that it would be easier to undertake anthropometric measurements and communicate advice on obesity once they provide a clear explanation about the link between obesity and oral health:

“*We have to explain to them that obesity and the weight have a big impact on oral health. They’re not comfortable, especially with the initial appointment, but probably by second or third appointment, they would be more comfortable. You need a very good explanation on the first appointment to let them know what these measurements are for*.”

#### 3.2.2. Barriers for Children and Their Families

**Limited health awareness:** The OHCPs reported limited health awareness among families as a barrier to undertaking obesity prevention and management in a dental setting. Some participants suggested that only when a child has developed dental caries, the parents are less reluctant to discuss about other concerns such as obesity. A few OHCPs also indicated the need for patient education to raise awareness about the link between oral and general health:

“*A lot of parents are unaware how much or how little sugar they need for their kids to develop decay. You have to wait for a hole to form before you can make the point. If patient education is not there, they’re going to think, why are you talking about my kids’ weight when we’re here for the teeth. When the knowledge improves, it becomes easier for us to include it in our clinics. If patients just think oral health is separate to general health, which a lot of them do, it becomes tricky*.”

Some OHCPs emphasised the lack of awareness among families concerning the role of diet in developing dental caries and obesity. One participant outlined that most parents consider fruits and juice as being healthy, disregarding the presence of sugar, which does contribute to both conditions:

“*When I explain to the parents about how much sugar there is in natural juice, they’re horrified because they don’t associate fruit with sugar. They associate fruit as something really healthy for you. So, they need to be educated that an orange might have at least two teaspoons of natural sugar in there. And then they realise that juice is not such a great thing to have all the time. A glass of juice is almost as bad as a glass of coke*.”

**Guilt due to difficult conversations:** Most OHCPs perceived the guilt arising among parents as a result of difficult conversations about their children’s weight as a barrier to obesity prevention and management. Some participants expressed that most parents would not be comfortable receiving advice about obesity in a dental setting:

“*From experience, I can’t imagine patients or even parents of the children accepting that, if I was to tell one of my patient’s moms, your daughter is a little bit overweight, she should start exercising. I don’t think that the public view oral and overall health as together. I’m giving them that kind of advice, they will say what’s that got to do with the teeth*.”

Some OHCPs emphasised that families would feel guilty and unhappy to hear that their child is overweight, especially coming from a dental practitioner. Despite the families’ reluctance, one participant suggested that they do bring up the issue of obesity and provide referrals:

“*Telling a family that your child is above the average weight, they might be a little bit guilty and not happy. But we can say, if you want any more help, we can refer you to a dietician. But whether the parents will take that that’s another thing. And we’re not a doctor, you’re only a dentist or dental therapist, you only work with the teeth. Why should you be telling me about the weight? So, that’s the other issue.*”

**Access and costs of health care:** A majority of the OHCPs perceived the costs as well as access to dental and health care services as significant barriers for the families. One participant also highlighted the need for Medicare to cover the costs of services provided by OHCPs as well as specialists to make these services more accessible for the families:

“*We do get a lot of cases where the parents explain to us that they really cannot afford it. I’ve had this connection with parents over the years where a lot of them are struggling to pay for dental and different services for their children. So, if it was covered through Medicare, it would make things a lot easier and more accessible*.”

A few OHCPs also mentioned that they had previously sought approvals for families experiencing financial hardship. Nonetheless, despite affordability issues, some of these families did not have any other option than to go to a private practice:

“*We recommend all these little things that we can do for them. But in the end, if we are unsuccessful, the only option would be to tell them they have to go privately. If the parent provides with a lot of information in regard to their financial hardship, then I have gone out of my way to try and seek special approval for that. And sometimes it has gone through if the management says that the parents are experiencing financial hardship*.”

**Uncertainty on scope in dental settings:** Most OHCPs emphasised that families frequently expressed uncertainty on the scope of obesity prevention and management in a dental setting. The families raised questions about the scope, resulting in the OHCPs being reluctant to undertake anthropometric measurements or provide weight counselling:

“*I don’t feel confident enough with their defiance, if the patients are uncooperative, if they show prejudice, like we’re oral health clinician and it’s not our area. None of your business, just do my teeth, you can have patients like that. So, patients that have that mentality, we can’t push this because there’s no point. It’s just like a waste of time as well.*”

Some OHCPs mentioned that those families who were comfortable with them undertaking anthropometric measurements may find the procedure unnecessary in a dental setting, depending on their child’s behaviour:

“*The obstacles we did face was when the child didn’t want to stand on the scale or didn’t want to have their height measured. The parent would really try for the child to participate. But if it’s unsuccessful, they say is this necessary? And we just say don’t worry about this.*”

**Fear of dental environment for children:** Most OHCPs indicated that the fear of dental environments among children did pose as a barrier to undertaking obesity screening:

“*A lot of the occasions it was because the child has never been in such a setting. It’s like a first exposure maybe to a dental setting or maybe they’ve just got some trauma from previous experience. So, they’ve come in thinking we’re going to give them an injection*.”

Nonetheless, the OHCPs mentioned they would utilise the first appointment to establish a rapport with a child who is uncooperative, allowing them to undertake obesity screening and counselling in the subsequent appointments:

“*So, if a child is uncooperative, I usually recommend another appointment, and use that appointment just to establish some rapport with the child, try to make it as friendly as possible, not worry on the treatment side. If the child is really anxious, crying in the chair, I would probably bring them back for another appointment*.”

#### 3.2.3. Enablers to Healthy Weight Status

**Education and training:** All OHCPs indicated that they had not received any structural training in anthropometric measurement or weight-counselling as part of their university curriculum or Continuing Professional Development (CPD). All participants emphasised the need for education and training on diet and obesity, including weight and height measurements, BMI for age charting, and BMI interpretation:

“*I studied over 30 years ago, so we had nothing about diet and obesity, it was just diet and teeth. If we have training, and if we’re given information on where these people can seek help, that will help us a big deal to help them as well. Extra training in the BMI thing, and the average height, weights, and then if you have a face-to-face, you can ask questions*.”

All OHCPs were receptive to workshops integrating an anthropometric training program, and courses on nutrition, obesity, and overall health. Most OHCPs were enthusiastic about courses and trainings that award CPD points from a professional organisation:

“*I think offering courses with CPD points. I know there’s already some courses running about diet and nutrition, but that’s more about gut health. So, this could be something that’s made into those courses currently running, so then talk about overall health and obesity.*”

**Diet diaries as a tool:** The OHCPs in this study were not using diet diaries for diet assessment. Nonetheless, a few participants indicated that using a diet diary as a risk assessment tool would assist them in comprehensively assessing the children’s diet in terms of quality and quantity, as well as identifying at-risk children in their practice:

“*We can recommend a 24-h diet diary or a weekend diary to note down cariogenic or obesogenic foods and even protective foods like cheese and dairy, to look at the quantity and quality of the foods that the kids are having. This would help us assess their diet in a comprehensive manner*.”

A few OHCPs elucidated the potential use of diet diaries to track the positive changes in children’s diet between appointments as per the dietary advice they have provided:

“*Small changes all the time, like maintaining a diet diary and using it as a tool to track changes could be useful. Some kids are eating lollies every day after school, so I would say, can you try and change that to maybe every third day after school. I would explain why it’s healthier. And when they come back for the next recall, we can have a look at the diet diary and see if it works*.”

**Family-centred approaches to achieve goals:** The OHCPs highlighted the need to utilise family-centred approaches to achieve goals concerning obesity prevention and management. A few OHCPs also suggested that weekly or fortnightly reminder messages could be useful:

“*The parents would like to get the children off iPads and watching TV. But now you see a child and then you don’t see the child for another three or six months. You get on a waiting list and in between the time, every advice to promote something that you’ve given just go waste. If we send them reminder messages every week or fortnight, that would be great.*”

Some OHCPs mentioned that as parents may be defensive when they are told to change their behaviour, it is essential to point out that any changes would be for the child’s health benefit:

“*Behaviour is very hard to change. Parents, because their mind is set, so I normally talk about children’s health and oral health. As soon as I mention it’s for child’s health benefit, they normally listen. Rather than that, if you just say you have to do this and you have to do that, they don’t like being told as they are parents.*”

**Patient education resources:** The OHCPs emphasised the need for visually appealing and interactive educational brochures for the parents, along with the contact information for various services available to them, which would facilitate referrals to specialists:

“*Simple pictorial pamphlets for the parents that’s got the average heights, weight and waist measurement, and where we can mark where your child fits in, and where you can get some extra help. We can say you can access these people, and I’m happy to make a referral.*”

A few OHCPs also highlighted the need for multimedia resources that can be televised in the waiting areas of dental practices. The participants expressed that the educational resources would inform the visiting families about their children’s co-vulnerability to dental caries and obesity, and simple effective strategies that can implemented to address weight issues:

“*Instead of having the normal channels on a TV, we can have oral health information screened in the waiting room, even mentioning the link between caries and obesity. Although the patients are here for a toothache or a preventative service, they’re getting more information. They’re likely seek information from other health professionals or even ask when they are in the dental setting.*”

**School- and government-run programs:** The OHCPs demonstrated a keen interest in learning about providing referrals to school- and government-run health promotion programs. Some OHCPs believed that particularly for young children with weight issues, school-run programs focused on promoting activity could be useful:

“*Maybe we could refer to the schools if there’s some connection with the canteens and maybe the teachers as well. So, if there is a child that has specific concerns, they can engage them in more exercise or develop a program for them. I guess schools for the young kids, but if there are some older ones that are interested in some exercise outside of school, then maybe contacting those centres and working with them as well.*”

Some OHCPs expressed that they informed parents about several government-run health promotion programs such as Go4Fun, Healthy Children Initiative, and Munch & Move, as well as school holiday programs that they were aware of:

“*I’ve told some parents that there’s a healthy children’s website under the government listings. You can look at that and there’s another school holiday programme, where they help with the exercising with the kids.*”

### 3.3. Theme 3: Role of Oral Health Professionals in Promoting Healthy Weight Status

#### 3.3.1. Role of Oral Health Professionals: Oral Health Therapists, Dental Therapists, Dentists, Paediatric Dentists

All OHCPs recognised the importance and inclusion of obesity screening, prevention, and management, along with providing dietary advice, as part of their routine scope of practice:

“*I don’t think it’s hard to take someone’s height, weight and waist measurements. Advising patients about their diet, that works well into a dental appointment. Part of the appointment is diet advice and you talk about sticky foods, carbs, and sugars. So, I think it ties in well because it’s part of our role, its within our scope. It’s what we’re supposed to do, promote healthy eating and obesity prevention or management*.”

Most OHCPs acknowledged that every specialty within the dental profession has an equal role and responsibility in providing dietary and obesity-related advice:

“*I think every oral health professional should have some idea of providing advice about healthy eating, because for a patient, they don’t really see a GP often unless they get sick. So, I think all oral health professionals should give some advice in terms of diet and weight.*”

#### 3.3.2. Bridging the Gap in Public–Private Dental Services

The OHCPs raised several issues within private practice, and the differences between public and private dental services. OHCPs working in private practices suggested that patients paying out-of-pocket for dental services might not be willing to cover the additional expenses for obesity screening and management. However, OHCPs working in the public sector stated that this issue would not arise, as the expenses would be covered:

“*In a private practice, where the parent had to pay out of their pocket for obesity screening and management, they’re more less likely to use it. Whereas, in the public, if Medicare was to cover it for free, then it would be an easy option for them*”.

Some OHCPs stated that referral pathways in public are easier than private practices, and emphasised the need for a specialist referral pathway and a database for specialists to engage with OHCPs to bridge the gap between public and private dental services:

“*Public is a lot easier to refer because they have a dietitian or nutritionist who works there. In private practice, nutritionists and dietitians aren’t closely engaged with oral health professionals. Bridging that gap between public and private, the nutritionist, the dietician referrals, is a huge factor. So, creating a database where nutritionists and dietitians can engage with oral health professionals might be a good idea.*”

## 4. Discussion

This study highlights OHCPs’ unique position to screen and manage other chronic conditions aside from those related to oral health. To our knowledge, this study is novel in Australia, as it explores OHCPs’ current practices, the enablers and barriers to practices, and intentions related to providing childhood obesity screening and management, which is usually considered peripheral to their primary clinical role. The findings corroborate the results of earlier studies that have shown OHCPs’ receptiveness to weight screening and counselling within their routine scope of practice [23,25,41,42,43,44,45]. In summary, this study provides preliminary insight for opportunities to re-orient the oral health workforce in obesity prevention and management.

Overall, the OHCPs acknowledged the link between oral health and systemic health and were well-versed with the concept of CRFA in health promotion [21]. Participants also identified diet and low socioeconomic status as the risk factors that are common between obesity and dental caries. This is a relatively positive finding [46,47] considering inconsistent literature of the relationship between obesity and dental caries [7,15]. For example, earlier studies found that most dental hygienists had some understanding of risk factors for systemic diseases, but were still unsure of the relationship between obesity and dental decay, and therefore were not willing to provide obesity-related counselling unless the link between the two conditions was well-established [42,46,47]. The present study is relatively unique, since it explores OHCPs’ knowledge on the link between obesity and oral health specifically, as well as their common risk factors [41,42,43,44]. Moreover, in the future, it would be useful to assess the variability in the degree of knowledge related to the oral–systemic health link and common risk factors across the wider OHCPs stream, that is, paediatric dentists, general dentists, oral therapists, dental therapists, dental hygienists, and dental assistants.

At present, the participating OHCPs were undertaking height and weight measurements, which could be because the OHCPs were provided training while they were a part of the HSHK study. However, they were not undertaking BMI charting and interpretation or obesity risk assessment for their paediatric patients. Earlier studies primarily based in the United States (US) and United Kingdom (UK) reported that OHCPs were providing anthropometric and weight-counselling services in their routine scope of practice, but they were very limited [25,41,42,43,44,45,46]. Nonetheless, the OHCPs in the present study did undertake diet assessment, patient education, and referrals. However, the diet assessment was informal (not using any diet diary or food frequency questionnaire (FFQ)) and mainly focused on children’s meals, sugar-sweetened beverages, and fruit juices intake. Dietary advice to children and families was primarily focused on limiting the intake and frequency of sugars and snacks and avoiding sticky sugar-containing foods, considering this is an integral part of their dental curriculum. However, the OHCPs were not well-versed with the recommendations stated in the Australian Dietary Guidelines [48], which would enable and equip them to better assess the children’s dietary behaviours and subsequently educate families on adopting healthy diets outside the scope of oral health. Moreover, consistent with previous literature [25,41,42,44,45], OHCPs in this study were not referring their patients to specialists due to the lack of knowledge regarding the referral pathway process, and only children who they believed to have weight issues were referred to either a GP or dietitian. Interestingly, the present study identified a gap in the referral pathway process amongst OHCPs working in public and private dental services, respectively. A recent review found that the possible reasons behind why most OHCPs may not be or are not able to undertake obesity prevention and management initiatives, unless patients inquired, included limited knowledge and training on anthropometric measurements, BMI interpretation, a lack of referral pathways, and time constraints [24].

None of the participating OHCPs received any structural training in anthropometric measurement (including weight and height measurement, BMI for age charting, and BMI interpretation) or weight-counselling as part of their university curriculum or CPD. These findings are consistent with other studies conducted overseas [22,26,42,43,44,46]. Furthermore, OHCPs reported having limited knowledge on the Australian Dietary Guidelines [48,49], which evidently inform health professionals on healthy and unhealthy food types, what age foods should be introduced, and in what quantity and frequency these should be consumed according to the age of the child. Educating OHCPs on the Australian Dietary Guidelines and training them in recording diet diaries or FFQs will enable them to effectively undertake dietary assessments [50] in a consistent way and disseminate dietary advice to children and parents outside the scope of sugars and oral health.

Additional barriers were time constraints on the part of OHCPs and uncertainty towards scope of obesity screening and counselling during a dental appointment on the part of both OHCPs and attending families. These perceptions have been echoed in previous studies [25,41,42,44,46]. It may be perceived that individuals attending a dental appointment would prefer to only receive their dental treatment, therefore an additional anthropometric assessment and/or weight-counselling may be a lower priority for both the patient and dental service provider, especially within the limited dental appointment time (usually 40 min). One possible strategy to address the issue of time constraints could be to train the receptionists and dental assistants at dental practices to undertake children’s weight and height measurements within the period between their arrival at the practice and prior to seeing the dental care provider. This workforce resource has potential, and its usefulness can be further explored [51]. Furthermore, regarding time management, an interventional trial [26] showed that dental hygienists were able to collect information on a child’s obesity-related risk factors (such as food intake, meal habits, physical activity, and screen time), measure the child’s height and weight and calculate BMI-for-age percentile in less than 40 min. Studies have reported that most participating parents were receptive to discussing their child’s weight with their dental practitioner [23,27], which highlights the potential of overcoming this barrier through better communication and rapport building between families and their dental care providers.

An additional obstacle reported by OHCPs was reluctancy to initiate a discussion around children’s weight status with parents, primarily due to a fear of offending or appearing judgemental. Similar barriers have been reported in other studies [25,42,43,46], which identified that a lack of knowledge about obesity and its causes and lack of training in weight counselling were limiting factors to initiate a discussion. One way of overcoming this limitation is to educate and train OHPCs in anticipatory guidance and using effective ways to initiate discussions with parents related to their child’s weight in a compassionate and sensitive manner to avoid stigmatisation. Evidence-based curricula to educate and train clinicians to identify children at risk of obesity and promote healthy weight amongst paediatric patients and their families are currently available [52,53]. These can be adapted and formally integrated within the Australian undergraduate dental curriculum or CPD vocational training to empower and give confidence to OHCPs to initiate the difficult weight-related conversations without causing children or parents to feel guilty.

Certain enablers were also identified which reflect positivity and scope of development. OHCPs were very receptive to training or workshops; thus, it is vital to develop and integrate an anthropometric training program and award CPD points from a professional organisation such as the Australian Dental Association (ADA) or Royal Australasian College of Dental Surgeons (RACDS). Research suggests that health care professionals are inclined to value the CPD activities [54], since they are obligated to meet the minimum CPD requirements to maintain their vocational registration [55]. The OHCPs can be trained on the well-recognised approach in health promotion known as 5 A’s, that is, Ask, Assess, Advise, Assist, and Arrange [53], to screen and manage childhood obesity. The proposed CPD training courses should cater to the learning needs of OHCPs on anthropometric measurement, growth charting, BMI interpretation, risk assessment, counselling, and initiating referrals to specialists. Since a majority of participating OHCPs identified time constraints as one of the important barriers for uptake of free resources, it will be appropriate to design flexible training programs, preferably delivered as online evening sessions or in a blended learning mode [56]. Moreover, anthropometric assessment training can be incorporated within the undergraduate and postgraduate curriculum to ensure new and post graduate dental providers have the basic knowledge and are confident in providing obesity screening and weight counselling. As reported in the present study, this aspect of training is currently limited across Australian dental institutions and hence needs to be urgently integrated, considering it is already being implemented overseas [57]. This aligns well with the integrated health care approach to disease prevention, which is being widely promoted to tackle the rising burden of chronic diseases in Australia [34,58,59].

The OHCPs emphasised the need for visually appealing educational brochures and multimedia resources (that can be televised in the waiting areas of dental practices) for visiting families to inform them about their children’s co-vulnerability to obesity and oral diseases, the common risk factors, and what simple effective strategies they can implement to address their children’s weight issues. Furthermore, designing a short risk assessment tool comprising diet diaries or FFQ, lifestyle behaviour questions, and height and weight charts would be useful to assist OHCPs to identify at-risk children in their practice. Use of such short anthropometric assessment tools has been shown to be successful in identifying patients at risk of overweight or obesity in dental practice [26]. The effectiveness of this tool will be augmented if a formal specialist referral pathway beyond a dietitian or a GP is developed and made accessible to both public and private OHCPs, respectively [45]. Interestingly, besides the specialist referral pathways, OHCPs showed a keen interest in having first-hand knowledge of referrals to local school- and government-run health promotion programs. Nonetheless, some OHCPs were already informing parents about several government-run health promotion programs such as Go4Fun [60], Healthy Children Initiative [61], and Munch & Move [62]. Referring to and promoting such children-centred initiatives might establish a family centred approach and possibly overcome the dilemma associated with specialist health care coverage costs.

Lastly, the OHCPs acknowledged the importance and inclusion of obesity screening and management as part of their routine scope of practice. They also recognised that every specialty within the dental profession has an equal role and responsibility in providing dietary and obesity-related services. These views and intentions are consistent with findings from a recent systematic review that concluded that most OHCPs based in developed countries such as the US and UK accepted and were very receptive towards their role in assisting paediatric patients at risk of obesity [24]. Hence, this study reinforces the usability and effectiveness of the emerging integrated health care approach in Australia [34,58,59].

Another interesting finding was that OHCPs, particularly those working in the private sector, stated that patients paying out-of-pocket for dental services might not be willing to spend additional money for obesity screening services. This situation will not arise in the public sector, considering the expenses will be covered by government. A possible solution might be to establish a third-party reimbursement system to cover the medical screening services or to develop a state-level specialist referral pathway to bridge the gap in public–private dental services. Encouragingly, research indicates that patients are willing to pay for medical screening in dental settings [63,64]. Hence, future research to assess the cost-effectiveness of obesity screening and management services within dental settings would be crucial.

### 4.1. Limitations

This study has certain limitations. The findings might not be a true representative of the perceptions and practices of OHCPs across NSW or Australia, primarily because of the smaller number of participants (*n* = 15) working in different regions of Sydney and across public and private sectors. However, considering there is no other research on this topic in Australia, these findings are valuable and can serve as a source for future research to corroborate the findings. Another possible limitation might be that the study relied on self-reporting rather than observed behaviours of OHCPs. Nonetheless, it is worth considering that the results are consistent with similar studies and those in other healthcare settings, hence they can have potential implications for dental professionals worldwide.

### 4.2. Implication for Practice, Policy and Future Research

The study findings suggests that OHCPs are interested in but do not receive training to promote obesity screening, prevention, and management in their routine of practice. There is an important opportunity to identify the problem of overweight and obesity in at-risk populations and improve their overall health. Several measures are required to design and incorporate anthropometric measurement to improve the current clinical dental practice. Such improvement requires an evidence-based model of care, including capacity building of OHCPs, development of resources (anthropometric assessment methods, risk assessment tools, anticipatory guidance techniques, and lifestyle-related modification counselling), and identifying a clear specialist referral pathway. Such a model of care could address the unmet needs of children with co-existing oral health problems and obesity. Development of a validated short risk assessment tool for use by OHCPs would be useful to appropriately identify paediatric patients at risk of overweight or obesity. An inter-professional team approach, including dental professionals, nutritionists, and dietitians, should be promoted and integrated into the current primary health care setup. Future research should (1) evaluate the cost effectiveness of employing OHCPs in providing obesity screening and management; (2) examine OHCPs’ perceptions about their conversations and stigma related to patients’ weight status, screen time, physical activity, and overall lifestyle; (3) explore the efficacy of dental receptionists in measuring weight and height and BMI charting for paediatric patients; and (4) explore parental perceptions of OHCPs’ role in obesity screening and management within dental practice settings. It is also important to consider that since the completion of interviews in this study, NSW Health has developed a guideline on growth assessment and dietary advice for children in public oral health services [65,66]. The document outlines training of OHCPs for the successful implementation of routine weight status assessment and management, along with the provision of brief advice and referral pathways to healthy lifestyle programs. Hence, future research to assess the implementation of the routine growth assessment and advice into clinical care would be valuable.

## 5. Conclusions

This study provides novel insights into the perceptions of OHCPs on current practices, the enablers and barriers to practices, and intentions related to providing childhood obesity screening and management, which is usually considered peripheral to their primary clinical role. The OHCPs acknowledged the link between oral health and systemic health and identified sub-optimal diet and low socioeconomic status as the common risk factors between obesity and oral diseases. Although the OHCPs were not undertaking any BMI charting and interpretation or obesity risk assessment for their paediatric patients, they did undertake height and weight measurements, diet assessment, patient education, and referrals. However, the OHCPs were not well-versed with the recommendations stated in the Australian Dietary Guidelines, which would enable and equip them to better assess the children’s dietary behaviours. The present study also identified a gap in the referral pathway process amongst OHCPs working in public and private dental services. Additional barriers were identified as time constraints on the part of OHCPs and uncertainty towards scope of obesity screening and counselling during a dental appointment. However, the OHCPs acknowledged the importance and inclusion of obesity screening and management as part of their routine scope of practice. A need for an evidence-based model of care was identified, including capacity building of OHCPs, development of resources such as anthropometric assessment methods and lifestyle-related modification counselling, and identification of a clear specialist referral pathway. Overall, this study provides an insight into opportunities to re-orient the existing allied or primary health care work force involved in obesity prevention and management.

## Figures and Tables

**Table 1 nutrients-14-01809-t001:** Themes and subthemes.

Themes	Subthemes
Theme 1: Obesity prevention and management in dental practice	Perceived link of oral health and general health
	Understanding of the common risk factors
	*Diet*
	*Socioeconomic status*
	Current prevention and management practices
	*Diet assessment*
	*Patient education*
	*Providing referrals to dietitians and general practitioners*
Theme 2: Barriers and enablers to obesity prevention and management in dental settings	Barriers for oral health professionals
	*Time*
	*Referral pathways*
	*Limited knowledge on dietary guidelines recommendations*
	*Limited knowledge on Body Mass Index (BMI), BMI for age chart, and obesity*
	*Scope of practice*
	*Communicating advice to children and families*
	Barriers for children and their families
	*Limited health awareness*
	*Guilt due to difficult conversations*
	*Access and costs of health care*
	*Uncertainty on scope in dental settings*
	*Fear of dental environment for children*
	Enablers to healthy weight status
	*Education and training*
	*Diet diaries as a tool*
	*Family-centred approaches to achieve goals*
	*Patient education resources*
	*School- and government-run programs*
Theme 3: Role of oral health professionals in promoting healthy weight status	Role of oral health professionals: oral health therapists, dental therapists, dentists, paediatric dentists
	Bridging the gap in public–private dental services

## Data Availability

The data presented in this study are not publicly available due to privacy.

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
