# Peer review of "Perceptions and Practices of Oral Health Care Professionals in Preventing and Managing Childhood Obesity"

_nutrients, 2022, doi:10.3390/nu14091809_

Round 1

Reviewer 1 Report

Authors correctly state that.....Overweight and obesity during childhood is associated with increased risk of  obesity in adulthood, as well as greater risk of developing chronic diseases such as cardiovascular diseases and diabetes at a younger age...but they should add also another important disease that is commonly present in overweight-obese children that is NAFLD, as evident in....

Clinical characteristics and risk factors of nonalcoholic fatty liver disease in children with obesity. BMC Pediatr 21, 122 (2021). https://doi.org/10.1186/s12887-021-02595-2.

The importance of detecting NAFLD in this setting relies upon the fact that there are many drugs on the pipeline that can cure this disease likely since very young age, as evident in....M, 02 Nov 2021, 126:154925

Author Response

Comment 1: Authors correctly state that.....Overweight and obesity during childhood is associated with increased risk of  obesity in adulthood, as well as greater risk of developing chronic diseases such as cardiovascular diseases and diabetes at a younger age...but they should add also another important disease that is commonly present in overweight-obese children that is NAFLD, as evident in....

Clinical characteristics and risk factors of nonalcoholic fatty liver disease in children with obesity. BMC Pediatr 21, 122 (2021). https://doi.org/10.1186/s12887-021-02595-2.

The importance of detecting NAFLD in this setting relies upon the fact that there are many drugs on the pipeline that can cure this disease likely since very young age, as evident in....Metabolism: Clinical and Experimental, 02 Nov 2021, 126:154925
DOI: 10.1016/j.metabol.2021.154925 PMID: 34740573 

Response 1: Thank you for the valuable suggestions. We have edited this line and included non-alcoholic fatty liver disease in the list of chronic diseases associated with childhood overweight and obesity. We have also included the suggested reference. Please refer to Page 1 Line 45. Since the focus of this paper is on childhood overweight and obesity and dental caries, including specific information on detection and treatment of NAFLD may be beyond the scope of this paper.

Reviewer 2 Report

This is a review of the manuscript “Perceptions and practices of oral health care professionals in 2 preventing and managing childhood obesity” submitted to Nutrients.

Introduction

The second and third sentence are a little repetitive. They not only refer similar things but also start in the same way. Can you rephrase it?

Methods

The authors state that all OHCPs involved in the HSHK study were invited; 15 were interviewed for this study. Can you please indicate how many invitations were sent (and the participation rate)?

Results

The results are presented in a clear way. The separation between different topics makes the content easy to follow and find.

The authors found some differences considering the public and private practice. In wonder if some differences could also be found regarding the area of exercise (e.g., dental therapists, general dentists, dental assistants).

Discussion

This section is well written and provides important points. The implication for practice, policy and future research was particularly welcomed. An inter-professional team would be the perfect approach worldwide but I do not know how this can be done, excluding in the public health care where all the services are available in the same space. Nevertheless, I guess it depends of the country.

Overall, the manuscript is really well-written, informative and on-point. Given the (country) innovation of the theme, the conclusions may have valuable implications for future childhood obesity and dental caries prevention research. It was a pleasure to read it. Kind regards,

DR

Author Response

Comment 1: This is a review of the manuscript “Perceptions and practices of oral health care professionals in 2 preventing and managing childhood obesity” submitted to Nutrients. Introduction- The second and third sentence are a little repetitive. They not only refer similar things but also start in the same way. Can you rephrase it?

Response 1: Thank you for the suggestion. We have rephrased both the sentences as suggested. Please refer to Page 1-2 Line 43-55.

Comment 2: Methods- The authors state that all OHCPs involved in the HSHK study were invited; 15 were interviewed for this study. Can you please indicate how many invitations were sent (and the participation rate)?

Response 2: Thank you for pointing this out. A total of 15 OHCPs involved in the HSHK study were invited and all 15 OHCPs completed the interviews, resulting in 100% participation rate. We have clarified this in the methods. Please refer to Page 3 Line 141-147.

Comment 3: Results- The results are presented in a clear way. The separation between different topics makes the content easy to follow and find. The authors found some differences considering the public and private practice. In wonder if some differences could also be found regarding the area of exercise (e.g., dental therapists, general dentists, dental assistants).

Response 3: Thank you for the positive feedback on the results. In this study, differences were observed among oral health care professionals (OHCPs) working in public and private practice. However, no differences were observed regarding the area of exercise. Most OHCPs (including dental therapists, oral health therapists, general dentists, and paediatric dentists) acknowledged that every specialty within the dental profession has an equal role and responsibility in providing dietary and obesity-related advice. We did not interview dental assistants. Please refer to Page 12 Section 3.3.1.

Comment 4: Discussion- This section is well written and provides important points. The implication for practice, policy and future research was particularly welcomed. An inter-professional team would be the perfect approach worldwide but I do not know how this can be done, excluding in the public health care where all the services are available in the same space. Nevertheless, I guess it depends of the country.

Response 4: Thank you for the positive feedback on the discussion and implication for practice, policy and future research. We agree that an inter-professional team would be the perfect approach. In the discussion, we have emphasised the need to bridge the gap in public-private dental service by developing a state-level specialist referral pathway (Page 16 Line 748). We have also emphasised the need for a formal specialist referral pathway beyond a dietitian or a GP, which is made accessible to both public and private OHCPs (Page 15 Line 724-727).

Comment 5: Overall, the manuscript is really well-written, informative and on-point. Given the (country) innovation of the theme, the conclusions may have valuable implications for future childhood obesity and dental caries prevention research. It was a pleasure to read it. 

Response 5: Thank you for your positive feedback on our manuscript. We agree that the conclusions may have valuable implications for future childhood obesity and dental caries prevention research.